# Development of the Digital Assessment of Precise Physical Activity (DAPPA) Tool for Older Adults

**DOI:** 10.3390/ijerph17217949

**Published:** 2020-10-29

**Authors:** Rosie Essery, James Denison-Day, Elisabeth Grey, Emma Priestley, Katherine Bradbury, Nanette Mutrie, Max J. Western

**Affiliations:** 1Centre for Clinical and Community Applications of Health Psychology (CCCAHP), University of Southampton, Southampton SO17 1BJ, UK; r.a.essery@soton.ac.uk (R.E.); j.l.Denison-Day@soton.ac.uk (J.D.-D.); emmaclare.priestley@hotmail.co.uk (E.P.); kjb1e08@soton.ac.uk (K.B.); 2Centre for Motivation and Health Behaviour Change, Department for Health, University of Bath, Bath BA2 7AY, UK; ebg21@bath.ac.uk; 3Physical Activity for Health Research Centre (PAHRC), Institute for Sport, Physical Education and Health Sciences, University of Edinburgh, Edinburgh EH8 8AQ, UK; nmutrie@exseed.ed.ac.uk

**Keywords:** physical activity, older adults, measurement, digital health

## Abstract

Physical activity (PA) is central to maintaining health and wellbeing as we age. Valid, reliable measurement tools are vital for understanding, and evaluating PA. There are limited options for comprehensively, accurately and affordably measuring older adults’ PA at scale at present. We aimed to develop a digital PA measurement tool specifically for adults aged 65+ using a person-based approach. We collated evidence from target users, field experts and the relevant literature to learn how older adults comprehend PA and would accept a digital tool. Findings suggest that older adults’ PA is often integrated into their daily life activities and that commonly applied terminology (e.g., moderate and vigorous) can be difficult to interpret. We also found that there is increasing familiarity with digital platforms amongst older adults, and that technological simplicity is valued. These findings informed the development of a digital tool that asks users to report their activities across key PA domains and dimensions from the previous 7-days. Users found the tool easy to navigate and comprehensive in terms of activity reporting. However, real-world usability testing revealed that users struggled with seven-day recall. Further work will address the identified issues, including creating a single-day reporting option, before commencing work to validate this new tool.

## 1. Introduction

Physical activity (PA) is widely recognised as a key constituent of healthy ageing given its protective positive association with cardiovascular health, cognitive health, physical frailty and chronic disease [1,2,3,4,5]. Moreover, PA affords older adults the ability to maintain their independence, engage with their communities and maintain a high quality of life into their later years [6]. The UK Chief Medical Officer (CMO), American Centre for Disease Control (CDC) and the World Health Organisation (WHO) are among the many global organisations who advocate that, to achieve these benefits, older adults should aim to: accumulate 150 min of at least moderate-intensity PA per week; reduce their sedentary time; undertake activities targeting strength and balance on at least two days per week [7,8,9]. Unfortunately, older adults tend to be the least active sector of the population: recent population data from the UK suggest that only 12% of people over 65 years old and 6% of people over 75 years old meet both the guidelines for aerobic activity and strength and balance [10]. In order to help preserve health and wellbeing in this ever-growing proportion of society, there remains a need to fully understand and increase older adults’ PA [11].

The precise measurement of PA is essential for improving our understanding of, and ability to help optimise, relevant behaviours. Measurement of both the quality and quantity of individuals’ PA allows better understanding of its relationship to health and wellbeing, through identifying individuals or groups who are inactive, by providing feedback to individuals on the appropriateness of their current PA behaviour, and by evaluating the success of interventions or initiatives aimed at increasing PA [12,13,14]. The gold standard of energy expenditure measurement is indirect calorimetry (IC) measured by metabolic chambers or carts, or the doubly labelled water (DLW) technique. However, these methods are expensive and unfeasible to use at scale, and do not capture other health-harnessing dimensions of PA [15]. For use in the field, current commonly used PA measures can broadly be categorised as device-based or self-report. Device-based (sometimes referred to as objective) measures such as accelerometers have been heralded as a superior method for quantifying PA in terms of moderate- and vigorous-intensity PA (MVPA) and sedentary behaviours [16], but can require great effort on the part of the processing researcher or practitioner to retrieve, handle and analyse their data [17]. Furthermore, they provide little contextual information about the type of PA and remain costly—often prohibitively so—for use in large-cohort studies or intervention trials [18].

Alternatively, self-report measures are typically paper-based tools and benefit from being affordable for use in large studies, and facilitate the collection of qualitative and quantitative information about the type of PA. However, these approaches are criticised for their reliance on participants’ recall, which could be influenced by limitations to memory and social desirability bias [18]. Furthermore, such measures often focus on assessing MVPA, whilst neglecting other key dimensions of PA for older adults [19]. Sustained MVPA is often unobtainable for older adults for multiple physical (e.g., cognitive, mobility impairment) or psychosocial reasons (e.g., low self-efficacy, loneliness) [18]. There is mounting evidence that replacing sedentary behaviours with light activity, strength, balance and short bouts of movement/sedentary breaks has enormous merit for maintaining health, wellbeing and independence in older adults [20]. Accordingly, it is vital that a tool to measure older adults’ PA captures these dimensions too. Indeed, a recent systematic review concluded that many existing self-report measures do not pay sufficient attention to content validity to allow comprehensive assessment of older adults’ PA [21]. A final, but important, limitation of existing self-report tools relates to evidence that they are often difficult to interpret and respond to (i.e., low face-validity), using terminology and/or response formats that are not well understood or clearly explained [22,23,24]. Developing a method to assess older adults’ PA that overcomes the respective limitations of contemporary tools would be highly advantageous for both research and practice [25].

A promising approach may be to harness the functions of digital technology to deliver a tool that facilitates the self-reporting of all dimensions of older adults’ PA with an option for the provision of feedback. There are several reasons to think this may be a valuable avenue to explore. Firstly, older adults are increasingly more prevalent users of digital technologies including personal computers, tablets and smartphones [26,27], meaning that this is an ever-more familiar platform for them to engage with. Additionally, the potential functionality afforded by the online (versus paper) delivery of a PA assessment tool (e.g., interactive components, visual representation, on-screen prompts) may plausibly help to overcome some of the limitations of existing self-report measures—e.g., complex response formats. Furthermore, the ease with which digital tools can be shared and engaged with by users [28] could provide an easy and cost-effective method for collecting data on a large scale.

To ensure the development of a suitable digital assessment tool, we first need to understand how acceptable a technology-based tool would be to older adults, and explore their understandings of terminology and conceptualisation of PA. As such, it seems vital that the approach to developing such a tool should have a central emphasis on seeking the views and experiences of those it will be designed for. A useful framework for guiding a rigorous user engagement process is the Person-Based Approach (PBA) to intervention development [29]. The PBA comprises two core elements: (1) conducting in-depth qualitative research from the outset of the research process to understand the needs, preferences and life-context of intended users, and (2) the formulation of Guiding Principles to underpin the development of the relevant tool [29]. It is an approach that, to date, has primarily been successfully applied to the development of behaviour change interventions across a range of health contexts [30]. However, the PBA’s core elements can be usefully applied in the current context.

The primary aim of this study was to employ Person-Based approach to guide the development of a tool for the ‘Digital Assessment of Precise Physical Activity’ (DAPPA) amongst adults aged 65 and over. Specifically, we aimed to develop a tool that accurately captures the key dimensions of older adults’ PA, is easy to interpret and use, and has the ability to provide brief feedback to users on their current behaviour. This paper outlines the development of the tool as well as key findings from that process.

## 2. Materials and Methods

We employed a Person-Based Approach [29] to developing the DAPPA tool, triangulating our findings with evidence gathered from a range of additional sources. This occurred in two interrelated phases—planning and optimisation. These phases were conducted in an iterative manner (see Figure 1) and the contributing elements of each are described further below. Ethical approval was obtained from the Research Ethics Approval Committee (REACH), University of Bath, Reference: EP 18/19 019.

### 2.1. Planning Phase

The planning phase of the process aimed to collect evidence from a range of sources to inform what should be the most important objectives and features for the tool, culminating in the development of ‘Guiding Principles’. Guiding Principles are a way of structuring the underpinning knowledge and understanding of the target users and relevant behaviours in order to maximise the acceptability, engagement and effectiveness of the tool being developed [29]. There were three main components of this phase: reviewing the existing literature, qualitative interviews with target users of the tool, and an expert survey—each is described further below. Findings from each of these processes, along with the Guiding Principles, are reported in the subsequent results section.

#### 2.1.1. Reviewing the Literature

Early in the planning process, we conducted rapid scoping reviews [31] of two key areas of the literature relevant to the development of the tool: (1) articles relating to existing self-report measures of PA—especially those for older adults; and (2) the qualitative literature about older adults’ experiences of PA. This review work aimed to gain deeper insight into:Common problematic features or characteristics of existing self-report PA measures;Any issues specific to PA measurement in older adults;The meaning and importance of PA to older adults; andOlder adults’ use and understanding of language and terminology relating to PA.

For both areas of the literature, we conducted searches in Medline, Embase, CINAH, PsychINFO, Cochrane database and Web of Science between February and April 2019. Initial searches for articles relating to measurement of older adults’ PA returned large numbers (multiple thousands) of matches, many of which were not relevant for title screening. Accordingly, we restricted this search to match terms in the title and abstract only and limited to review articles only. The resulting matches were still numerous and far more relevant to our aims. Database searches were conducted by EP and RE. Matching records were imported into Rayyan QCRI [32] where duplicates were removed and screening decisions were made and recorded. For the ‘measures of PA’ literature, EP screened all records and RE screened 60% to check for agreement. For the ‘older adults’ experiences of PA’ literature, RE and MW independently screened all articles. In both cases, agreement was high (>90%) and any differences in decision were discussed and agreement reached. After screening, we extracted and collated information about: specific measures examined in the reviews; any strengths and limitations discussed; key characteristics of the measure, such as whether it was designed for a specific population or measured specific dimensions of PA. From the qualitative experiences of the PA literature, we extracted data from each of the included studies about key themes arising in the analysis.

#### 2.1.2. Focus Group and Qualitative Interviews

Alongside the literature reviewing, we also conducted one focus group (*n* = 5) and ten one-to-one qualitative interviews with older adults to explore their experiences and understandings of various aspects of PA. Specifically, this primary qualitative work sought to gain a better understanding of:What PA means to older adults and how they think and talk about it;Their experiences of measuring their own PA;What their perceptions are about the role of technology in PA; andWhat types of activity they choose to engage in and why.

Participants were recruited through identification via the University of Southampton’s Psychology Volunteer Participant database. Potential participants matching eligibility criteria (aged 65 and above, willing and able to access the internet, and who perceived themselves to be physically capable of at least some PA (i.e., not currently injured or disabled) were sent an email invite to participate in either a focus group or a one-to-one semi-structured interview at a suitable location for all participants. RE and JDD facilitated the focus group and qualitative interviews were conducted by JDD. The same interview schedule was used in both and covered the topics described above. The second phase of the interview provided participants with a two existing commonly used self-report PA measures (the International Physical Activity Questionnaire-Elderly (IPAQ-E) [33] and Community Healthy Activities Model Program for Seniors (CHAMPS) [34]) and asked them to spend a few minutes completing these before discussing their impressions of these measures. All interviews were audio-recorded and transcribed verbatim. Data were analysed and organised into themes according to the principles of thematic analysis [35].

#### 2.1.3. Expert Survey

In parallel with the other development activities, we designed a brief online qualitative survey to obtain the views of professionals working with older adults in a variety of PA-related settings. In particular, we sought input from academic researchers focussing on PA, physiotherapists, occupational therapists and those working in the leisure industry, such as personal trainers and activity class leaders. The aim was to understand the views of professionals who frequently use self-report PA measures. In particular, we were keen to hear about:What they deemed to be important aspects of PA for older adults;Their perceptions of common barriers to older adults’ PA;Any issues they experienced with commonly used self-report measures; andPerspectives on optimum measurement of older adults’ PA.

The brief survey consisted of one multiple choice and five short free-text responses and could be completed by anyone who had the URL to access it. Members of the research team shared this URL with relevant contacts in their network. It was also shared via Twitter by MW and RE, who encouraged retweeting by their networks—particularly amongst those with relevant expertise. Local leisure centres were also contacted to share with their staff, who work with older adults. Responses to the survey were collected automatically and downloaded from the survey software. The free text responses were descriptively coded and summarised, drawing on areas of consensus but also seeking to present the range of views and experiences expressed.

### 2.2. Optimisation Phase

The planning phase culminated in the development of ‘Guiding Principles’ that shaped the creation of a fully functioning prototype tool. Each Guiding Principle consisted of a design objective, capturing a behavioural need, or a challenge identified as relevant and important to older adults in the context of measuring and obtaining feedback on their PA, and key features of the measurement tool that aimed to address each of these objectives [29]. These guiding principles underpinned the development of the prototype tool. In the following optimisation phase, we then conducted think-aloud interviews to obtain feedback on all aspects of the prototype tool to inform the changes required to maximise its suitability, acceptability and usability for intended users. In the secondary part of this phase, we ran a preliminary validation study to assess the construct validity of the prototype tool. This compared data collected by the revised version of the tool with that from a commonly used device-based measurement of PA, and three paper-based self-report tools. Participants in this validation study were asked to provide feedback on the usability and acceptability of the tool during a brief telephone interview at the end of their participation. Each of these elements of optimisation is discussed in further detail below.

#### 2.2.1. Think-Aloud Interviews

Ten think-aloud interview participants were recruited from those participants willing to be re-contacted from the development phase qualitative interviews and focus group (*n* = 15). During each think-aloud interview participants were asked to use the prototype DAPPA tool and to share their immediate thoughts and reactions to all aspects of tools (content, presentation and functionality) with the researcher present (JDD). Users were primarily encouraged to offer their own thoughts on each page of the tool but were also prompted for additional information where necessary (e.g., “*What are you thinking now?”, “What do you understand by that message”, “What made you decide to click on that button?)*. These interviews were audio-recorded and transcribed verbatim.

The data were then analysed to rapidly identify changes that could maximise the acceptability, usability and engagement with the tool. To do this, we began by collating all positive and negative comments relating to specific elements of the tool into a ‘table of changes’. We discussed the frequency and relative importance of these positive and negative comments within the team and coded possible targets for change by deciding whether any amendment was likely to enhance the tool’s accuracy, usability and acceptability. For example, we considered whether several participants provided the same feedback, if the potential change aligned with our guiding principles, and whether any of the evidence we had collected reinforced that the change could make the tool more accurate, easy to use, or engaging for older adults [36]. We prioritised changes that met one or more of these criteria—for example, an issue raised by multiple participants that also aligned with one of the tool’s guiding principles would be considered high-priority. Low-priority changes were only implemented if they were relatively simple and non-controversial. This analysis was conducted alongside ongoing think-aloud interviews to allow iterative modification of content prior to the next interview where changes were non-controversial. More complex changes were left until the end to gather the maximum amount of data. Once it seemed that no further important changes were required, we considered that data saturation had been reached [36].

#### 2.2.2. Usability Study

Once adaptations were made based on the think-aloud interview data, a new set of participants (*n* = 20) were recruited to use the tool for a two-week period. The objectives of this study were:To investigate the user facing and technical functionality of the DAPPA tool when used in a real-world context;To evaluate the acceptability of the DAPPA tool to users and identify further improvements; and, if the data permitted,Explore the preliminary validity of DAPPA and compare its assessment with existing device based and self-report measures.

Participants for the usability study attended a set-up session in which they signed informed consent and completed a demographics questionnaire. They were given an ActivPAL [37] PA monitor (Criterion) and three self-report questionnaires: International Physical Activity Questionnaire-older adults short form (IPAQ-E, [33]); Physical Activity Scale for Elders (PASE, [38]); and the Godin Leisure Time Questionnaire (LTEQ, [39]), against which to compare the data from the DAPPA survey. The ActivPAL device, which has been validated for assessing sedentary and PA behaviour [40,41] was placed in a waterproof latex sheath and fitted to the midpoint of the left anterior thigh of the participant using Tegaderm tape. Participants were instructed to keep the device on for 14 consecutive days.

The DAPPA tool was designed to capture PA behavior over the previous 7 days. As such, participants were instructed to complete the DAPPA tool and first set of questionnaires after eight days, and the second set after 15 days (so that the data in each case related to the seven previous days full ActivPAL wear). An email prompt containing a link to the DAPPA survey and the participants’ unique log-in was sent to participants on these days, along with a reminder to complete the three self-report questionnaires. After completion of the second weeks’ data collection, participants posted their surveys and the ActivPAL device back to the research team in a prepaid envelope. Participants who indicated that they would be happy to be interviewed then underwent an audio-recorded telephone interview that asked them about their experiences of using the DAPPA tool.

## 3. Results

### 3.1. Planning Phase

Figure 2 summarises the key findings from the literature review [18,22,23,24,42,43,44,45,46,47,48,49,50,51,52,53,54,55,56,57,58], qualitative data and expert survey that contributed to the development of the Guiding Principles and, subsequently, to the prototype intervention itself. Additional detail about the results of each activity is provided in the relevant sections below.

#### 3.1.1. Reviewing the Literature

In the ‘measurement of PA’ area of the literature, our searches initially identified approximately 1200 records matching our search terms. After the removal of duplicates (approximately 900) and title and abstract screening, we were left with 48 articles for full-text review, of which 30 were deemed relevant to inform the guiding principles. In the ‘older adults’ experiences of PA’ area of the literature, initial searches matched approximately 1000 records. After the removal of duplicates (approximately 600) and title and abstract screening, we were left with 70 articles for full-text review, of which 34 were retained to inform development of the guiding principles. Key evidence collated from these rapid reviews of the literature is summarised in Figure 2, with example references from the review evidence.

#### 3.1.2. Focus Group and Qualitative Interviews

Table 1 provides the participant characteristics of individuals who took part in either the focus group (*n* = 5) or a one-to-one interview during the development phase (*n* = 10).

A summary of the key findings arising from the thematic analysis and illustrative quotes are provided in Table 2 below.

#### 3.1.3. Expert Survey

Nine individuals responded to the expert survey, all of whom identified themselves as academic researchers working within the field of older adults’ PA. All respondents indicated familiarity with at least one existing self-report PA measure—most commonly the IPAQ (short or long version), PASE or CHAMPS. The majority reported concerns about the accuracy of data obtained from existing self-report measures citing recall issues, over-reporting and difficulty for users in interpreting items. Several suggested that simpler wording and more visual elements may facilitate response to such measures. A large proportion of respondents cited strength and balance training as especially important for older adults’ PA. With regards to frequency, duration and intensity, they tended to agree that frequent bouts of activity, even of a short duration and light intensity, are likely to be beneficial and achievable for most older adults. Respondents’ views about common barriers to PA amongst older adults largely coincided with those identified from the literature reviews and from the qualitative interviews. They suggested that illness, pain and mobility-related issues often prevented engagement in PA, as well as other commitments and responsibilities taking up time, and a lack of access to appropriate facilities or resources.

### 3.2. Guiding Principles

The findings from the planning phase described above were triangulated and informed the Guiding Principles, as outlined in Table 3 below. These guiding principles underpinned the development of the prototype tool described in the following section.

### 3.3. The DAPPA Tool

The DAPPA tool was developed as a responsive web application enabling users to access and use the tool from any device with an internet connection, such as desktop computers, tablets and mobile phones. The tool was given a user-facing name ‘My Activity Diary’. The tool consists of three sections: user administration, the activity diary, and feedback.

The *user administration* section consists of a registration/login function, user admin (updating account information and passwords) and the homepage. During the initial registration process, users provide their age, gender, height and weight. Following registration, or login for returning users, users are presented with the homepage, which has buttons to access the tool, change their password, update their personal details (age, gender, weight, height) and log out. 

In the tool, users are initially asked to insert an approximate time they wake up and go to sleep each day. They are then shown an instruction page before reaching the Activity Diary. Users are advised that they will be required to report their activity for the preceding seven days. The Activity Diary page starts with a heading that informs the user which day they are filling in activities for, including the day, date and how many days ago this was (up to seven). Completing the activity diary then involves selecting activities completed from 83 available activities (Appendix A) chosen for inclusion on the basis of the primary interviews and the literature. These activities are organised into four domains based on the way older adults described their activity behaviour: ‘Home and Garden’—activities conducted in and around the home; ‘Out and About’—activities that are part of daily life routines or getting from place to place; ‘Sport and Exercise’—activities performed intentionally with the purpose of keeping fit and active; and ‘Social and Leisure’—activities that are secondary to meeting with others or hobbies/pastime activities.

Once the type of activity is selected, users are asked to report other key dimensions of each activity, e.g., how long they engaged in this activity for and, optionally, whether they did it in the morning, afternoon or evening. Once completed, the chosen activity turns green to show that it has been successfully entered. Users are able to deselect, edit and re-select activities. They can also move backwards and forwards between days to allow them to make changes should they remember new activities or need to correct mistakes. After completing the diary for the most recent day, users are then prompted to submit their diary.

An additional feature evaluated in the present study was the use of a *feedback* page that informed user about their PA behaviour for the week comparative to the UK Chief Medical Officer’s guidelines for older adults [7]. This feedback page, designed to be distinct from the measurement tool itself, includes both text and visual information about: the minutes of light, moderate and vigorous exercise completed; how many times strength and balance activities were done; average daily sleep; average daily sedentary time. A tailored feedback message was also included based on their proximity to achieving the 150 min of moderate-to-vigorous PA and two times per week strength and balance activity. For example, users who achieve less than 50% of the recommended targets, see a message including the following feedback:

‘*Any amount of physical activity has benefits even if it’s only a few minutes here and there or lighter levels of activity. Being physically active is one of the best ways to make sure you can keep doing the things you enjoy, stay healthy and strong, and feel good. To get the most benefit, it is recommended that adults do at least 150 min of moderate activity a week. If higher levels of activity are a bit too much at the moment, doing light activity is still great. Light activities help to break up the amount of time you spend sitting or lying, and can help build you up to more energetic activities when you’re ready, so keep going with those activities!*’

Figure 3 below shows screenshots of key pages of the tool before and after feedback from think-aloud interviews.

A final, non-user-facing, feature of the DAPPA tool is the data extraction function. Data collected by the activity diary include total minutes of light, moderate and vigorous activity, classified using the Ainsworth (2011) compendium of physical activity Metabolic Equivalent of Task (METs) [59]. The number of strength and balance activities, walking time and sedentary time can also be extracted. The age, height and weight data input by the user are used to estimate resting metabolic rate, which, in turn, enables an assessment of energy expenditure based on reported activity. This information is also broken down by day and, where given by the user, morning, afternoon and evening. For each 24-h period, time that was not allocated to sleep or specific physical activities is assumed to be sedentary time.

### 3.4. Optimisation Phase

Table 4 summarises participant characteristics of those participating in both the think-aloud interviews and the usability study elements of the optimisation phase.

#### 3.4.1. Think Aloud Interviews

Ten participants took part in one think-aloud interview each. The interviews ranged from 21 to 53 min, with a mean duration of 33 min. In general, participants were positive about the prototype tool from the early interviews, with most finding it relatively easy to navigate and understand. Several commented that they found it better than the paper versions they had worked with in the previous phase of work, in terms of being easier to complete and edit as they went along and the sense that it would provide a more accurate picture of their activity. It was also considered relatively comprehensive in terms of the activities to choose from. However, participants did raise a number of issues that highlighted aspects of the tool requiring further work in order to optimise its utility and usability. Table 5 shows a summarised excerpt from the table of changes’ analysis, and outlines some of these key issues, and how they were addressed.

#### 3.4.2. Usability Study

Of the 19 individuals who provided written consent to start the usability study, 17 provided complete ActivPAL data (in two cases, a fixing failure meant that the device was removed) and 18 provided complete questionnaire data. This study marked the first test of the DAPPA database under study conditions. Unfortunately, owing to unforeseen issues linking the PHP hypertext pre-processor (PHP) database with the University of Bath servers, data from the DAPPA tool were only retrievable for nine participants, rendering the preliminary validation futile. Nonetheless, participants were able to use the tool, and ten of the recruited participants (*n* = 5 female, mean age = 71 years (SD = 4 years), range = 67–77 years) agreed to take part in the telephone evaluation interviews.

##### Evaluation Interviews

Participants who completed evaluation interviews voiced both positive and negative perceptions of the DAPPA tool. In general, participants reported that they found the tool’s interface easy to use and navigate, although several would have liked more initial instruction on what activities need to be reported and are available to choose in the tool:


*“…there were other things, like cooking I suppose… I didn’t class as an active- you know, sort of general household stuff, bit of gardening I overlooked until I caught up with that, but I think if I’d actually had the list of possible activities in front of me before I started, I would have been better prepared.” *
(DPA004)

Related to this, some reported disliking the drop-down activity menus and would have preferred to see all the activity options in a single screen, which may have prevented them missing activities, particularly non-structured exercise activities.

The main criticism of the tool, reported by all interview participants, was the weekly completion; having to recall, in detail, all their activities from the past 7 days was difficult, and participants felt that it resulted in them giving inaccurate reports:


*“I couldn’t remember what I’d done five or six days ago exactly. I knew if I’d been to the shops, but not what I’d done in general... I felt I couldn’t be accurate enough.”*
(DPA002)

After completing the DAPPA tool for the first week, the majority of participants kept a written daily diary of their activity during the second week to help them complete the tool. Most participants felt that it would be quicker and enable more accurate reporting to complete the tool on a daily rather than weekly basis, although one participant thought that daily completion *“might become a bit of a chore”* (DPA011) and was happy to keep a written diary.

Participants appreciated the feedback given after completing the tool, with a couple mentioning that it had made them more aware of and motivated to minimise the amount of time they spend being sedentary. Others found the feedback reassuring *“that I am doing the right level of exercise for me”* (DPA007). While none reported being de-motivated or tempted to reduce their activity as a result of the feedback, one participant speculated that receiving feedback of higher than expected activity might *“make you complacent”* (DPA002). A couple of participants saw the value of the feedback for tracking progress and suggested that it would be helpful to see and compare results for individual days and at different points throughout the year,


*“getting your score for that day, so you can see the difference between the days, maybe you do some regular exercise or travel somewhere, or do something, you’d see the difference between those days, and days where you’re sort of sat around watching telly or reading the paper.”*
(DPA001)

## 4. Discussion

This paper has documented the systematic, person-based development of the DAPPA tool to measure older adults’ PA behaviour. This section outlines how key findings from our collated evidence relate to both tool development and the wider literature, and how we plan to take the development of the tool forward.

Several key messages arising from the study’s planning phase relate to the wider literature about older adults’ PA. Firstly, PA is thought of in very broad terms by older adults—not just within the traditional definitions of ‘exercise’ but as ‘movement’, and is frequently embedded in daily life activities. Indeed, a recent systematic review similarly concluded that older adults construe PA as being tied up in day-to-day life activities—particularly those that they value and enjoy—and stress the importance of recognising this for interventions aiming to increase such behaviours [60]. This understanding of how older adults perceive their PA behaviours played an important role in informing how the tool groups and presents activities for users to select from. Accordingly, the grouping of activities to select from in the tool is framed around the context in which activities occur, rather than in terms of activity intensity, as is often traditionally presented.

Relatedly, we found there to be widely held doubts about the utility of traditional terminology such as ‘moderate’ and ‘vigorous’ for helping older adults to understand and report PA. Our primary qualitative data collected from older adults and field experts both suggested such terminology to be problematic in terms of being difficult to interpret or judge, and apparent variation in individuals’ understandings of what these terms mean. For example, several participants’ comments suggested that they thought of terms such as ‘moderate’ and ‘vigorous’ in relation to the duration of an activity or distance travelled, rather than activity intensity. These findings were echoed in the evidence collated by the review of the literature. Indeed, this key message is reiterated in the findings of a recently published study exploring communication relating to PA behaviours amongst underserved communities (in which older adults were included). One of the study’s key conclusion was that the language used to communicate PA should be simple and jargon-free, avoiding terms such as ‘moderate’, ‘vigorous’, and ‘intensity’, that are often perceived as inaccessible or even intimidating [61].

A final key message concerns older adults’ perceptions of technology and its relation to PA behaviours. The collated evidence suggested that older adults have a growing familiarity with technologies such as computers, smartphones and tablets, and many are relatively willing and able to use these on at least a basic level. This aligns with national reports demonstrating older adults to be rapidly increasing users and owners of digital technologies and services [27]. Several of our participants recognised the potential value of wearable technology in terms of motivation and self-monitoring of PA behaviour, even if they had little interest in it for their own use. We took these views as a promising indication that a digital measurement tool should be at least *accessible* to many older adults. Furthermore, by involving them throughout the early development process, as advocated by recent recommendations about using technology to support older adults PA [62], we hope to have ensured that it is also usable, acceptable and engaging.

Following the iterative optimisation of the DAPPA tool, we obtained promising feedback to suggest that it is largely accessible and easy to-use with indications from some users that they feel it characterised their activity more accurately than their previous experience of paper self-report measures. However, the real-world usability study highlighted that the seven-day reporting format was deemed problematic by all users, placing too much demand on recall. There were also some indications that users were unsure about what activities should or should not be reported, leading to lack of reporting in some cases. Finally, there were issues with the technical data collection aspect of the tool, evidenced by the data loss in the real-world usability study. In terms of developing the tool further, we would prioritise addressing these problems before attempting to comprehensively validate the tool.

To address difficulties with the seven-day recall element of the tool, we plan to create a version that allows users to select how many days they wish to retrospectively report activity for. This would offer the flexibility of reporting activity a day at a time, or being able to report blocks of days’ activity at once if preferred. A recent study evaluating the relative performance of seven-day and single-day reporting of PA amongst young active adults concluded that the shorter recall periods may improve the measurement quality of the PA questionnaire [63]. Given that older adults are more susceptible to difficulties with memory and recall [18], this seems likely to be beneficial in this context. Moreover, a digital tool may alleviate issues concerning the feasibility and participant burden that a move towards a multi-sample, single-day PA measure might otherwise incur [63], as has been observed in the measurement of other health-related behaviours [64,65].

With regards to addressing problems with uncertainty about which activities need to be reported, we propose enhancing the instructional information at the start of the tool about the scope and nature of activities that should be reported. Whilst the tool does already provide instruction around this, the existing literature acknowledges the need for engaging with, and following, instructions as a potential limitation of diary or log-format measurement tools [21,42]. In trying to address this, we will attempt to make further use of the digital format to make these instructions more engaging—potentially by reducing the textual information that appears at one time, and by simplifying other aspects of the page to reduce any potential distraction from key messages [66].

Whilst this study collated evidence from a range of sources to inform development of the measurement tool, some limitations of the study should be noted. The number of people who contributed to the expert study was relatively small (*n* = 9), although not substantially smaller than samples used in expert consensus activities in other recent studies [67]. However, whilst we endeavoured to seek the views of a range of experts with this survey (health and sports practitioners and leisure industry professionals, etc.), we only succeeded in recruiting academic professionals. This may have precluded a wider range of experiences and views being recorded by the expert survey. Relatedly, it is possible that recruitment of our development phase participants from a volunteer participant pool risked selection bias in favour of those more interested in, and possibly knowledgeable about, the research process and topic. This may have affected the range of experiences and views we recorded. Whilst our sample characteristics did appear relatively varied, in further work we will aim to recruit from a broader range of sources and seek a maximum variation sample, particularly in terms of current physical activity levels, education status and age. In addition, although our initial intention had been to conduct a preliminary validation of the tool, the loss of data in the real world usability study meant that this was not possible. We believe the technical issues resulting in data loss to be a result of the incompatibility of the tool with the hosting institution’s servers. As such, there may not be a specific issue to address, but we will fully review and test the tool’s database to ensure confidence in its data collection and storage capabilities.

Once the database is fully operational and adjustments to the DAPPA tool itself have been implemented, the next logical step would be a thorough validation against criterion PA measures [25]. Whilst our priority for further development of the tool will be in its capacity as a measurement tool, a further potential avenue for development relates to the feedback element. In line with our planning phase findings about the motivational quality of feedback from PA-related technology, optimisation phase findings similarly suggested that feedback offered users useful insight into their current behaviours and could be valuable for keeping track of progress. In this respect, there may be scope for further optimisation of the activity diary inclusive of the feedback component. Such a tool could potentially act as a ‘light-touch’ PA intervention through offering a means of self-monitoring behaviour and receiving personalised feedback [68,69].

This study has created and provided preliminary evidence about the acceptability and content and face validity of a digital PA assessment tool for older adults. Whilst further work is still required to optimize and validate the tool, this represents an important step towards addressing the limitations of current PA assessment options [21], and consequently providing a better understanding of older adults’ physical activity behavior. Beyond the immediate relevance to the DAPPA tool, some observations arising from the development work will be of interest to practitioners and the wider research field. This includes the ambiguity in older adults’ interpretation of commonly used PA terminology, which may have implications for how PA recommendations are communicated.

In addition, this study has some important methodological implications. It has demonstrated how the person-based approach to intervention development can be successfully applied to the development of a measurement tool. The PBA’s application here provides evidence that its focus on in-depth understanding of target users and their behaviours, in combination with developing guiding principles, is transferable and relevant beyond the scope of intervention development. As well as offering a systematic development process, the PBA can provide deeper insight into the experiences and behaviours of those individuals at the centre of the process, making important contributions to the relevant literature. This evidence of the broader applicability of the PBA has important implications for any research seeking to develop a tool for users whose relevant experiences, life contexts and perceptions are poorly understood.

## 5. Conclusions

Using a systematic, person-based, development approach, we have created the Digital Assessment of Precise Physical Activity for older adults. The tool captures more than 80 activities of the past seven days and captures different domains (‘home and garden’, ‘out and about’, ‘sport and exercise’ and ‘social and leisure’) and dimensions (frequency, duration, type and intensity) of PA as well as sedentary behavior. Informed by the literature, field experts and target users, the DAPPA tool employs a simple to use, interactive online diary format. The tool aligns with older adults’ conceptualisations of PA behaviour and technological capabilities. Future work will refine the tool based on the learnings documented in this preliminary study, and validate it for use in the field.

## Figures and Tables

**Figure 1 ijerph-17-07949-f001:**
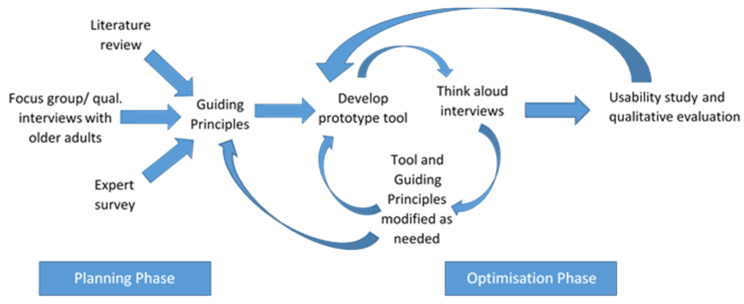
Overview of DAPPA tool development process.

**Figure 2 ijerph-17-07949-f002:**
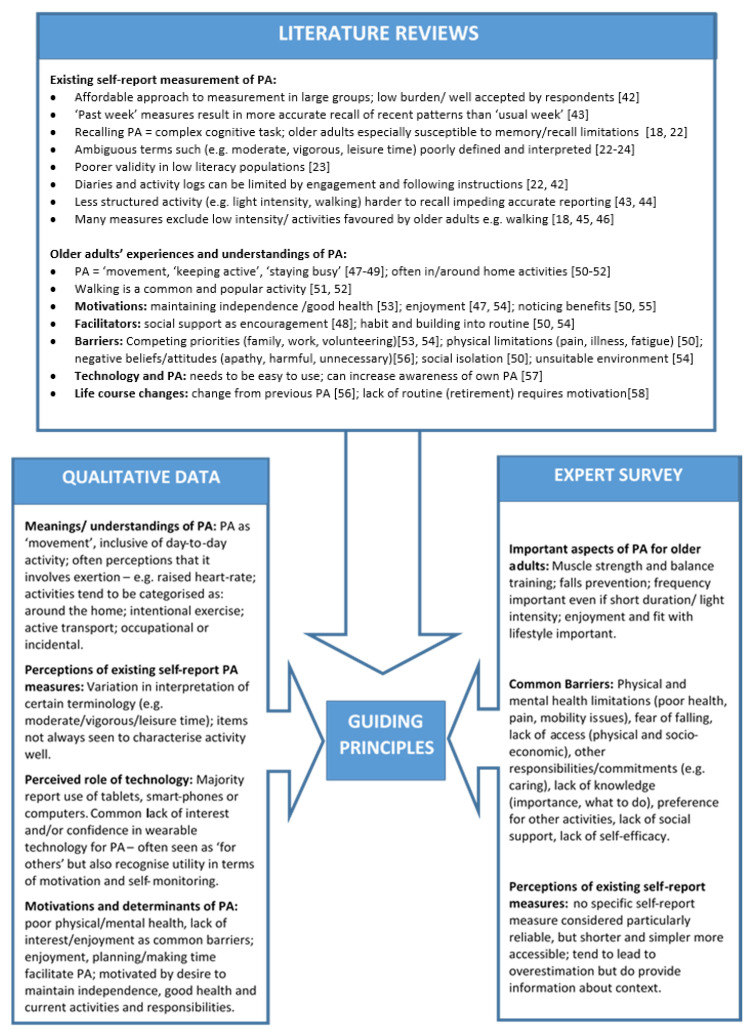
Key findings from planning phase.

**Figure 3 ijerph-17-07949-f003:**
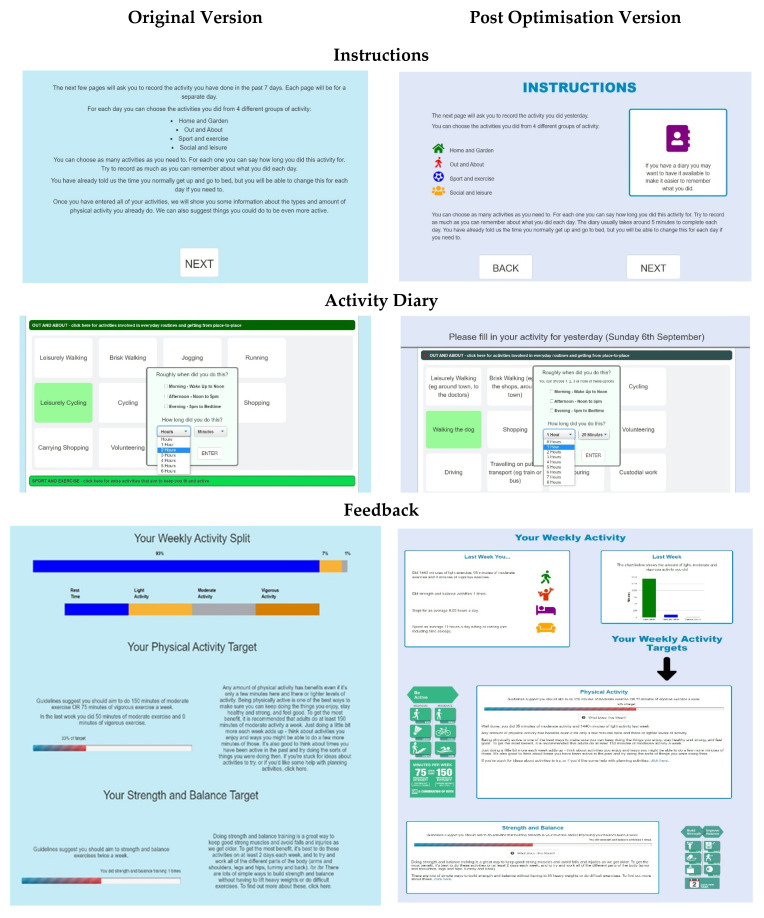
Screenshots from the My Activity Diary tool: original and post think aloud versions.

**Table 1 ijerph-17-07949-t001:** Characteristics of participants in the planning phase.

Characteristic	Planning Phase Participants (*n* = 15)
**Gender**	
Female	8 (53%)
Male	7 (47%)
**Age** (mean (SD) range)	74.3 (5.0) 68–83
65–74.9	8 (53%)
75+	7 (47%)
**Marital Status**	
Married/cohabiting	9 (60%)
Divorced/single/widowed	6 (40%)
**Education**	
Secondary or less	5 (33%)
Further education	5 (33%)
Higher education	5 (33%)
**Employment**	
Retired	14 (93%)
Part time employment	-
Unpaid volunteer	1 (7%)

Data reported as *n* (%) unless specified.

**Table 2 ijerph-17-07949-t002:** Key findings and illustrative quotes from qualitative data.

Key Findings	Participant Quotes
Physical Activity (PA) seen as ‘movement’; inclusive of day-to-day activity; involves exertion	“Yes, I think it’s about mobility, moving. I don’t think it’s particularly sports. It’s your activity that you do in your everyday life.” (Focus Group, Female participant)
	“Increased heartrate. Possibly sporting activities or even domestic, doing the gardening. Anything that gets you moving” (Focus Group, Male participant)
	“Movement and energy. It covers a whole range of things from day-to-day tasks of daily living to, at the other end, extreme sports, or anything in between such as walking, cycling. A whole range of stuff.” (P006, female, 76)
Variation in how existing common terminology interpreted; items not seen to capture one’s activity especially well	“Yeah, I’d describe it as slow walking or wandering rather than walking leisurely. Why not that? This is stationary but why not? Interesting. Why are they—just interested in why they separated it.” (P002, female, 69)
	“I just don’t feel that it captures… My problem is that I’m not a black and white person. […] I’ve got lots of fuzzy edges and the fuzzy edges don’t fit in with this and therefore you’re not going to capture a true story of people’s physical activity unless you look at the fuzzy edges.” (Focus Group, Female participant).
Majority familiar/ comfortable with computer/ tablet/ smartphone, but lack of interest and/or confidence in wearable technology—often thought of as more ‘for others’, but some recognition of utility in terms of motivation	“I’ve got friends that use [an activity monitor] and they use them religiously for about three months and never use them again. For me, what I saw of them is that you come quite manic about keep looking at their watch or whatever it is they were wearing, to see how many they’d done, and in the end they’d stand on the step on the spot and do steps, which is fine because they still use up the energy but I think either you walk or you don’t walk and knowing how far you’ve walked doesn’t really make much difference so for me personally, no, I wouldn’t bother with that.” (P001, female, 72)
	“Well, having an app that or a device that tells you how much activity you’ve undertaken is useful. I’ve got the Tom Tom watch that does the 10,000—well, it counts your steps. I appreciate they’re not that accurate but it’s a guide to how physically active you’ve been […] Yes, if I haven’t done many steps I think I should have done a bit more exercise today.” (P003, male, 68)
Common motivations and determinants (barriers and facilitators) of activity	“The hardest thing is to get people motivated and that’s for all of us in this room. We’ve all motivated ourselves somehow ‘cause we want to either live a bit longer, be a bit fitter and do things a lot longer than maybe our contemporaries.” (Focus Group, Male participant)
	“I used to love walking but I have got a lung condition and it limits me. I get very annoyed about it. I don’t think about it all the time but I’m quite a quick person and it’s when I try to do things too quickly I’m pulled back. Really I did play golf but I had to pack that up. I had to pack it up. I stick to the swimming now as the only thing I really do.” (P005, female, 71)
	“I’m very, very organised so I think to be honest I don’t think there’s anything that makes it difficult, maybe because I love it as well, I love the actual feeling of being out there on my bike. When I’m on the indoor bike I have a news stand in front with a book so I can read at the same time.” (P002, female, 69)

**Table 3 ijerph-17-07949-t003:** Guiding Principles for the development of the DAPPA tool.

Design Objective	Features
To minimise cognitive demand (e.g., recall, interpretation, clarity of instruction, ease of use)	Diary format to ask for reporting of specific days anchored with days and dates Reminder that it may be useful to consult a diary if kept Simple, jargon-free language Linear layout with simple login procedure
To present activity options in a meaningful, easy-to-interpret way	Activities to select from categories in line with how/where/when conducted Avoidance of asking people to report their activity in terms of ‘light, moderate, vigorous’ terminology as this is often open to interpretation
To allow easy reporting of wide range of activities across all dimensions important to older adults’ PA	Wide range of light, moderate, vigorous and strength/balance/flexibility dimension activities included for users to select their activities from Choices labelled according to specific activity Categories of activities organised by how/where/in what circumstances it might be performed (i.e., Home and Garden, Sports and Exercise, Out and About, Social and Leisure) Allows selection of activity, time of day and approximate duration rather than needing to type lots of information about activity type and duration
To increase older adults awareness/knowledge of their own PA	All activities assigned MET value that, alongside reported duration, facilitates calculation of minutes of exercise per week in different activity intensities. Presents brief (including visual) feedback about current activity levels in relation to national government guidance on PA for older adults Simple recommendations made tailored according to individual’s PA performance relative to national guidance and realistic for older adults in light of common barriers.

**Table 4 ijerph-17-07949-t004:** Characteristics of participants in the optimisation phase.

Characteristic	Optimisation Phase (*n* = 29)	
	Think Aloud (*n* = 10 ^a^)	Usability Study (*n* = 19)
**Gender**		
Female	4 (40%)	10 (53%)
Male	6 (60%)	
**Age** (mean (SD) range)	74.2 (5.4) 68–83	73.7 (5.5) 66–90
65-74.9	5 (50%)	10 (53%)
75+	5 (50%)	9 (47%)
**Marital Status**		
Married/cohabiting	7 (70%)	14 (74%)
Divorced/single/widowed	3 (30%)	5 (26%)
**Education**		
Secondary or less	3 (30%)	9 (47%)
Further education	4 (40%)	-
Higher education	3 (30%)	10 (53%)
**Employment**		
Retired	10 (100%)	17 (89%)
Part time employment	-	2 (11%)
Unpaid volunteer	-	-

Data reported as *n* (%) unless specified. ^a^ These were 10 of the 15 participants recruited into the planning phase.

**Table 5 ijerph-17-07949-t005:** Key issues arising from think-aloud interviews and changes implemented.

Section/Aspect of Tool	Summary of Issue Identified	Example	Change Implemented
**Activity choices and selection**	Some users mentioned looking for very specific examples of activities to report that they could not find.	“… it was mainly [going to] the doctor and the hospital because I have to go backwards and forwards several times there and, of course, you have to walk to the bus stop and then get on the bus and do that and that wasn’t covered. There was leisurely walking, but I consider leisurely walking going out for a leisurely walk, not your everyday things you have to do that you have to walk to.” (P001,female, 72)	Rather than add lots of additional separate activities that could become excessive to look through, such specific activities were added as examples in the relevant activity—e.g., within ‘Out and about’ section (activities outside the house that often include getting to and from places), we added walking to appointments as one of the examples in the ‘walking’ activity.
	Users sometimes missed reporting an activity/chose an alternative/related activity before later finding the appropriate one.	“No. Oh, I forgot about yoga. I do that most days but for about 10 min. Does that count as well? Right, okay. That’s interesting. Perhaps a reminder that people have to go right to the bottom because I didn’t go right to the bottom, and I do yoga every day. 10 min in the morning and 10 min in the evening. So, really, I’ve missed that out. Perhaps just a reminder. Go right to the bottom of the sheet.” (P002, female, 69)	Added brief text before the first page of the activity diary to encourage people to look through all categories of activities first before starting to report their activity to avoid missing out activities or choosing a less accurate alternative
**Duration/timing of activity**	Tool doesn’t allow user to indicate if an activity spanned multiple time periods—for example—late morning/early afternoon	“Yeah, ‘cause we were doing it in the morning and the afternoon, but not in the evening, and then I don’t seem to have an option to do morning and afternoon.” (P003, male, 68)	Modified reporting functionality so that users can tick all that apply from ‘Morning’, Afternoon’ and ‘Evening’ options.
**Recalling activity**	Several remarked that having their diary or calendar to hand would be useful in recalling what they did on specific days	“It might have been quite a good idea if I’d known I was gonna be thinking back for the week so that I had my […]. If I was going to do this then I would probably spend time thinking, oh yeah, and look at the calendar and think, oh, I did that, oh yeah, I did that. Because, as you get older you don’t remember.” (P009, female, 70)	Added a notice on the instructions page to recommend that users have a diary to hand if they use one, as it may aid recall of activities from preceding week.
**Clarity of instruction**	Users sometimes lost track of what day they were entering activity for	“Just for Thursday? (Interviewer: Yes). I thought it was for the whole seven days.” (P007, female, 83) “Yeah, perhaps that needs—does it actually state it and I’ve missed it? ‘Cause it had Thursday 11 July. Perhaps put Thursday 11 July on there—you know, the date on there as well.” (P002, female, 69)	Enhanced the ‘day/date/X days ago’ title at the top of each page to make this more prominent and also fixed this header so that it is visible regardless of how far down a page people scroll as a reminder.
	Some users were not aware that they could move between days during completion	“Ah the next day, can you go backwards? I jumped to the next day.” (P012, male, 77) “Oh I just remembered what I was doing last Thursday, I was coming back from France. Can we go back? Can I go back and change it?” (P011, male, 76)	Added detail about function of the back button by adding sentence next to it explaining that participants can return to previous days to amend or add to what they have already reported.

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
