# Peer review of "Development of the Digital Assessment of Precise Physical Activity (DAPPA) Tool for Older Adults"

_ijerph, 2020, doi:10.3390/ijerph17217949_

Round 1

Reviewer 1 Report

General comments

Dear authors,

Thank you for submitting this study to the IJERPH. In my opinion, the study is very interesting, important for the field and has high methodological quality. The measurement of physical activity is difficult and even though we have many available options, self-reports perform often rather poor (although this depends also on the quality of validation study). One problem is that the content validity and the psychological components associated with the measurement (e.g., how people understand and recall questions and activities) received only little attention in research.

This study provides essential evidence for the content validity of physical activity self-reports. The main goal of the study was to develop a digital tool which is accepted by and relevant for the population of older adults. The results of the study are extremely helpful for users and researchers who wish to either apply or further improve physical activity self-reports.

Following my evaluation of the manuscript, several minor points should be addressed. Thank you very much.

Abstract

Line 13: typo: ‚7‘

Line 14: It could be valuable to mention that not only there are limited options but in general little attention was paid on the concepts of content and face validity of PA self-report measures in older adults (e.g., doi: 10.1007/s40279-020-01268-x). My suggestion is also to address these two concepts in the introduction below since those are the ones covering many of your targeted issues, e.g., most important activities of older adults, understanding of questions etc.

Line 15: Perhaps consider rephrasing this section to a specific aim? e.g., the goal/aim of this study was to develop ….by using a person-based approach.

My suggestion would be to add the age cut off (> 65 years) to the abstract and the aim section of the introduction. This is an important characteristic of the sample.

Introduction

I suggest using ‘PA’ as an abbreviation for physical activity to improve the readability. This term is commonly used in the field (The term physical activity appears very often in the manuscript).

You may consider including very recent evidence from the Generation 100 study. Most of the available evidence for the health benefits of PA is observational, so rather than talking about ‘influence’ (e.g., line 31) it might be more appropriate to say something like ‘positively associated’ etc. (Generation 100 study doi: 10.1136/bmj.m3485). This study instead used an RCT design.

Line 48: In my opinion, IC is not the gold standard to measure PA. DLW is the gold standard only for total energy expenditure in free-living conditions. A gold standard for PA does not exist. However, different purposes have different ‘most appropriate tools’. For example, a pedometer for walking, DLW for total EE. Measuring PA perfectly would mean that there should be a gold standard which captures all the domains and dimensions of PA. I am not aware of such a tool (e.g. doi: 10.1186/s12966-016-0351-4 OR doi: 10.1016/S0140-6736(12)61876-5)

As mentioned above, you may add some information about the concepts of content and face validity. These concepts cover approx. the lines 61-74. Introducing those concepts would also be important for readers not directly working in the measurement field. In my opinion, only little attention was paid to those concepts although having good content validity is required in a first step, otherwise the evaluation of other validity and reliability parameters is useless.

Line 63: Good point. Indeed, most PA self-report measures for older adults neglect lower intensity activities, typically performed in that population. Often, the reason is because it is even more difficult to recall these activities. Based on this section (in which you thoroughly discuss the issue with the importance of light PA etc., e.g. line 69 too) I got the impression that your tool will also collect information of lower intensity activities, is that right? If not, you may wish to mitigate this section a bit?

Line 100: What are they ‘key domains’ for older adults? This was also not fully clear for me when I read the lines 59 to 74. Please clarify to me and perhaps modify the relevant sections in the introduction. For instance, I can think of traditional domains (household, occupation, leisure, transport) and dimensions (frequency, duration, intensity, type) of PA. MVPA is not really a domain, it is rather a dimension of PA, referring to the intensity of the behaviour. You can have MVPA in leisure time and transport for example.

Methods

Line 149: typo: ‘order’.

Line 149-151: It may increase the readability of the manuscript when you use bullet points for all the ‘sub-aims’ of the qualitative interviews (e.g., types of measures frequently utilised; any issues…). It is a lot of text. Such a presentation of the sub-aims can be applied to this section and perhaps to other sections as well (e.g., Line 170, Line 218 …). Maybe you can use and match those with headings presented in figure 2.

Line 154: ‘physically capable of at least some physical activity’ Please describe (at least to me) how and when this was assessed.

Line 164. Expert survey: How did you analyse this data? I can see you present the results within the main text in the result section. Maybe you can add some information to this section 2.1.3. as well, e.g., presenting frequencies or mean responses etc. Maybe it is worth considering presenting the results of this survey in a table or figure, instead of the text: How many of 9 people mentioned problems in the understanding or suggested that the frequency of PA is beneficial for older adults?

Line 186: You may consider using the correct term here, e.g., the goal was to assess the ‘construct validity’ of the first version of the tool.

Line 208: ‘We prioritised changes by their likelihood to address 208 one or more of these issues’. Could you please explain to me (and perhaps add the information to the text) what you mean with this?

Line 215: Please just clarify to me. The 20 people in the validation study are a ‘new’ sample and hence, have never seen the IPAQ etc. before? Why did you use PASE and LTEQ rather than CHAMPS, which was used in the other interviews, in the validation study? Also, why did you use the IPAQ-E here but the general IPAQ version for the interviews? Thank you for clarifying these issues to me.

Line 230: Perhaps write ‘wore the device for 14 consecutive days’? Some people may stop reading when they read 7 days because that is the most common data collection period for device-based measurements.

Line 231: Perhaps consider adding the information that the DAPPA tool was ultimately designed to capture the last 7 days. Although this is a result, I think it will help to understand why they were introduced to complete it twice after 7 and 14 days.

Line 235: Which other ‘other paper-based surveys’ do you mean? You may add this information here or refer to the respective method section of the manuscript.

Results

Line 240: You may consider referring to Figure 2 only once in the result section. For example, you can mention here that the main results of the three planning parts ‘literature reviews, qualitative data and expert survey’ are shown in figure 2; rather than referring to this figure at all occasions throughout the result section.

Table 1. IMD score. Is the deprivation score really necessary to present, as this score is never mentioned somewhere else in the manuscript? If you leave it in the table, I would describe the rationale for it and discuss the result as well.

Table 1. e.g. female/male, 65-74.9 etc. Please add the information how the data is presented (either in the table or footnote), just like it is done for the overall age (mean, SD). Something like n (%).

Line 282: This is a nice description of the guiding principles (e.g., design objectives; the idea is to derive a protocol for the design etc.) but earlier in the method section, line 182, only a very brief description was provided. I suggest adding the details of the principles (and all other steps) to the section of the manuscript where they are mentioned first (i.e., methods). I think it is not necessary to describe the steps again or even in more detail in the result section as this should be clear from the method section already.

Line 292. Please describe what the introduction (especially the recall period) by the tool for each participant was. At what time, the person should complete the diary etc., after 7 days? This was not clear for me when I started to read the paragraph. Also, wouldn’t it be more appropriate to call it a ‘recall’ rather than a diary? Most people would think that diaries are designed for a continuous monitoring (after each hour, in the evening).

Line 305: Would it be possible to list all 83 activities in an online appendix? That would be interesting for other users and researchers designing such tools.

Line 314: Could you describe somewhere how long it takes to fill in the diary for one person, on average? Is it very time consuming? This can also be discussed later, .e.g. with respect to the feasibility of your tool for larger applications.

Line 315: Typo ‘.’

Line 317: This feedback opportunity is interesting and valuable. Just clarify to me: Do they receive the feedback only once after the activities from the past 7 days were entered? Are there any consequences when you use your tool in intervention studies to measure change in PA by providing cues for changing the target behaviour? Can you de-activate the feedback in the tool?

Line 320. How many questions for sedentary time were there? Do you cover sedentary time in different domains, sitting at work, sitting during travel, in leisure time for tv viewing, or was it one overall question for daily sedentary time? I remember in the introduction you talked about the importance of different key domains etc.

Line 320: Can you provide an example for a tailored feedback message in the text?

Table 4: See comments Table 1.

Table 5: First row. In the updated diary, is there an option to add own/individual activities, if the activity is not listed? Did you consider this option?

Line 360: I appreciate that you describe the data loss transparently. Could you please explain why the ‘entire’ validation study was deemed as futile? Do you think it is worth showing the mean values, agreement in a descriptive, very brief, way for that 9 people?

Line 379: This is a good point. That was my impression too. Perhaps using the daily option is promising. It seems to be a lot of work when many participants have to record their activities ‘offline’ on a separate form and then transfer their answers to the digital tool.

Discussion

Line 448. I am not sure whether this study (nr 63) really ‘compared’ long and short version, e.g. by evaluating differences between them. Please clarify to me and/or in the text.

Line 470: What about the representativeness of your sample of older adults with respect to the general population? You may elaborate briefly on this topic. Do you have any concerns, as they have been invited using the psychology volunteer participant database; any selection issues with that? Perhaps they have a better understanding of health behaviours and concept? Was that an active or typically inactive sample based on the questionnaire results?

Line 475: Just a note. For the future validation study you may consider the accelerometer and, specifically the PASE, a questionnaire which seems to be very appropriate in older adults (doi: 10.1007/s40279-020-01268-x)

Conclusion

Line 500: I suggest adding again a very brief description of the tool to the conclusion section, something like that ‘the tool captures more than 80 activities of the past 7 days and covers different domains (the covered ones) and dimensions (the covered ones) of PA as well SB’.

Related to that could please explain to me again whether your tool covers now ‘all’ domains (i.e., leisure, household, occupation, transport) and dimensions (frequency, duration, type, intensity [light, moderate, vigorous) of PA (as well SB). You may provide details for that in the result/method section too. Readers often search whether the tool measures frequency of PA as well as which types and intensities are assessed. In the introduction I got the impression that your tool should also cover light intensity and the key domains of PA for older adults (what are the key domains again?)

Author Response

Many thanks for your helpful comments - our responses in bold italics.

General comments

Dear authors,

Thank you for submitting this study to the IJERPH. In my opinion, the study is very interesting, important for the field and has high methodological quality. The measurement of physical activity is difficult and even though we have many available options, self-reports perform often rather poor (although this depends also on the quality of validation study). One problem is that the content validity and the psychological components associated with the measurement (e.g., how people understand and recall questions and activities) received only little attention in research.

This study provides essential evidence for the content validity of physical activity self-reports. The main goal of the study was to develop a digital tool which is accepted by and relevant for the population of older adults. The results of the study are extremely helpful for users and researchers who wish to either apply or further improve physical activity self-reports.

Following my evaluation of the manuscript, several minor points should be addressed. Thank you very much.

We thank the reviewer for the close attention they have paid to our manuscript and the positive evaluation as to the rationale for the study and its rigor. 

Abstract

 Line 13: typo: ‚7‘

Thank you, correction made.

 Line 14: It could be valuable to mention that not only there are limited options but in general little attention was paid on the concepts of content and face validity of PA self-report measures in older adults (e.g., doi: 10.1007/s40279-020-01268-x). My suggestion is also to address these two concepts in the introduction below since those are the ones covering many of your targeted issues, e.g., most important activities of older adults, understanding of questions etc.

We thank the reviewer for brining our attention to this interesting and important systematic review. We have referenced the paper in the introduction and agree that discussing the problems faced by existing self-report tools in relation to a lack of attention on content and face validity helps articulate the rationale for our study (line 69-74). Due to the low word count afforded for the abstract we have opted not to explicitly mention these terms here.

Line 15: Perhaps consider rephrasing this section to a specific aim? e.g., the goal/aim of this study was to develop ….by using a person-based approach.

Thank you for this suggestion, we have rewritten this statement as an aim (line 15-16).

My suggestion would be to add the age cut off (> 65 years) to the abstract and the aim section of the introduction. This is an important characteristic of the sample.

We thank the reviewer for this suggestion and have added the age cut-off to the abstract (line 15) and the introduction (line 97-98).

Introduction

I suggest using ‘PA’ as an abbreviation for physical activity to improve the readability. This term is commonly used in the field (The term physical activity appears very often in the manuscript).

Thank you, we agree this improves the readability and have replaced physical activity with the abbreviation PA throughout the manuscript.

 You may consider including very recent evidence from the Generation 100 study. Most of the available evidence for the health benefits of PA is observational, so rather than talking about ‘influence’ (e.g., line 31) it might be more appropriate to say something like ‘positively associated’ etc. (Generation 100 study doi: 10.1136/bmj.m3485). This study instead used an RCT design.

Thank you for this suggestion. We have altered the text to suggest that PA is positively associated, citing the paper suggested (line 30-31).

 Line 48: In my opinion, IC is not the gold standard to measure PA. DLW is the gold standard only for total energy expenditure in free-living conditions. A gold standard for PA does not exist. However, different purposes have different ‘most appropriate tools’. For example, a pedometer for walking, DLW for total EE. Measuring PA perfectly would mean that there should be a gold standard which captures all the domains and dimensions of PA. I am not aware of such a tool (e.g. doi: 10.1186/s12966-016-0351-4 OR doi: 10.1016/S0140-6736(12)61876-5)

Thank you for this observation, we agree with this viewpoint and have changed the wording accordingly to state that IC and DLW only measure energy expenditure, and do not capture other health-harnessing dimensions of physical activity (Line 47-50)

 As mentioned above, you may add some information about the concepts of content and face validity. These concepts cover approx. the lines 61-74. Introducing those concepts would also be important for readers not directly working in the measurement field. In my opinion, only little attention was paid to those concepts although having good content validity is required in a first step, otherwise the evaluation of other validity and reliability parameters is useless.

Thank you for this suggestion, as noted in the corresponding point in the abstract we have now explicitly mentioned these concepts and included the suggested citation (line 67-72).

Line 63: Good point. Indeed, most PA self-report measures for older adults neglect lower intensity activities, typically performed in that population. Often, the reason is because it is even more difficult to recall these activities. Based on this section (in which you thoroughly discuss the issue with the importance of light PA etc., e.g. line 69 too) I got the impression that your tool will also collect information of lower intensity activities, is that right? If not, you may wish to mitigate this section a bit?

Thank you for this comment, our tool did seek to capture lower intensity activities of users and in addressing another of your comments, have added a list of all activities assessed by the DAPPA tool in a Supplementary File.

 Line 100: What are they ‘key domains’ for older adults? This was also not fully clear for me when I read the lines 59 to 74. Please clarify to me and perhaps modify the relevant sections in the introduction. For instance, I can think of traditional domains (household, occupation, leisure, transport) and dimensions (frequency, duration, intensity, type) of PA. MVPA is not really a domain, it is rather a dimension of PA, referring to the intensity of the behaviour. You can have MVPA in leisure time and transport for example.

We thank you for this observation and agree that we have misused the terminology. We have altered our use of the phrase domain to capture the context of an individuals PA (as you suggest) and dimension to capture the dose of PA (frequency, duration, intensity etc.) throughout the manuscript.

 Methods

 Line 149: typo: ‘order’.

Thank you, correction made (Line 147).

Line 149-151: It may increase the readability of the manuscript when you use bullet points for all the ‘sub-aims’ of the qualitative interviews (e.g., types of measures frequently utilised; any issues…). It is a lot of text. Such a presentation of the sub-aims can be applied to this section and perhaps to other sections as well (e.g., Line 170, Line 218 …). Maybe you can use and match those with headings presented in figure 2.

Thank you, we agree that the use of bullet points in these sections enhances the readability and have added them in (lines 125-128; 151-154; and 176-179). We have also reordered the presentation of these sub aims to more clearly map on to the findings presented in Figure 2 – the wording necessarily remains slightly different between the bullet point aims and Figure 2 for the purposes of readability/ presentation of the Figure (line 264).

Line 154: ‘physically capable of at least some physical activity’ Please describe (at least to me) how and when this was assessed.

Many thanks for highlighting that this was not clear. Physical capability was not formally assessed. We have clarified in the text that this criterion was based on older adults’ own perceptions of their ability to complete at least some (even if very minimal, i.e. not disabled or currently injured) physical activity (line 158,159). Our recruitment materials expressed that we did not need people to be very active at all as we were hoping to gather a range of experiences.

 Line 164. Expert survey: How did you analyse this data? I can see you present the results within the main text in the result section. Maybe you can add some information to this section 2.1.3. as well, e.g., presenting frequencies or mean responses etc. Maybe it is worth considering presenting the results of this survey in a table or figure, instead of the text: How many of 9 people mentioned problems in the understanding or suggested that the frequency of PA is beneficial for older adults?

Thank you for this query and suggestion. Apologies for any confusion – we now realise it was not clear that the expert survey collected predominantly open-ended qualitative data, rather than quantifiable short answer data (with the exception of one question which asked about self-report measures that respondents had experience of using). As such, we don’t feel it appropriate to quantify the responses, but have added text (line 188-189) to clarify this and provide further information about how this data was analysed.

Line 186: You may consider using the correct term here, e.g., the goal was to assess the ‘construct validity’ of the first version of the tool.

Thank you, we have added the suggested phrasing to include the accurate technical term (line 199).

 Line 208: ‘We prioritised changes by their likelihood to address one or more of these issues’. Could you please explain to me (and perhaps add the information to the text) what you mean with this?

Apologies for the unclear wording – we have rephrased to provide more clarity about what we mean in the manuscript (line 222-224)

For further clarification - in the person-based approach a pragmatic prioritisation of user comments is implemented to help ensure the changes made address key issues (i.e. those raised by multiple users or supported by theory) to prevent dramatic alterations being made based, for example, on the opinion of just a single user.

Line 215: Please just clarify to me. The 20 people in the validation study are a ‘new’ sample and hence, have never seen the IPAQ etc. before? Why did you use PASE and LTEQ rather than CHAMPS, which was used in the other interviews, in the validation study? Also, why did you use the IPAQ-E here but the general IPAQ version for the interviews? Thank you for clarifying these issues to me.

Thank you for your question. The 20 people in the validation study are indeed a new sample. We opted not to use CHAMPS in the validation study as the validated version would not have matched up with our DAPPA tool (and the other selected questionnaires) as it asks users to comment on average behaviour over 4 weeks, rather than just the past 7 days. We did however want to include it in the interviews as we felt it provided more breadth in terms of the style and format of existing questionnaires, and thus may have invited a wider range of thoughts and comments from participants on the strengths and limitations etc., of existing measures.

The IPAQ-E was actually used in both phases of the study given its reported validation for use in older adults. We thank you for bringing this error to our attention and have corrected this in the manuscript (line 164).

 Line 230: Perhaps write ‘wore the device for 14 consecutive days’? Some people may stop reading when they read 7 days because that is the most common data collection period for device-based measurements.

Thank you for the suggestion, we have amended the text as suggested to be clear the device was worn for 14-days (line 249).

 Line 231: Perhaps consider adding the information that the DAPPA tool was ultimately designed to capture the last 7 days. Although this is a result, I think it will help to understand why they were introduced to complete it twice after 7 and 14 days.

We agree that this will enhance the comprehension of this section of the methodology and have added text in to inform the reader that the DAPPA tool was designed to capture PA behaviour over the previous 7-days (Line 250).

 Line 235: Which other ‘other paper-based surveys’ do you mean? You may add this information here or refer to the respective method section of the manuscript.

Thank you for this observation. The text has been changed from ‘other paper-based surveys’ to ‘the three self-report questionnaires’ to enhance the clarity and keep the phrasing consistent with the previous paragraph (line 254-255).

Results

 Line 240: You may consider referring to Figure 2 only once in the result section. For example, you can mention here that the main results of the three planning parts ‘literature reviews, qualitative data and expert survey’ are shown in figure 2; rather than referring to this figure at all occasions throughout the result section.

Thank you for this suggestion. We have altered the text at the start of the results section to more explicitly describe the components found in Figure 2 (line 261) and have removed further reference to the figure.

Table 1. IMD score. Is the deprivation score really necessary to present, as this score is never mentioned somewhere else in the manuscript? If you leave it in the table, I would describe the rationale for it and discuss the result as well.

We originally included IMD data as we felt it might be a useful additional way of characterising the participants included in this development work. However, we agree that it does not feature in the discussion of our findings so have removed from this and subsequent tables.

Table 1. e.g. female/male, 65-74.9 etc. Please add the information how the data is presented (either in the table or footnote), just like it is done for the overall age (mean, SD). Something like n (%).

Thank you, we have added as a footnote the statement ‘Data reported as n(%) unless specified’ (line 280).

Line 282: This is a nice description of the guiding principles (e.g., design objectives; the idea is to derive a protocol for the design etc.) but earlier in the method section, line 182, only a very brief description was provided. I suggest adding the details of the principles (and all other steps) to the section of the manuscript where they are mentioned first (i.e., methods). I think it is not necessary to describe the steps again or even in more detail in the result section as this should be clear from the method section already.

Thank you for this suggestion, we have opted to move the text you describe to the methods section (line 192-196) and have added a new, briefer line that provides context to table 3 (Line 301-303)

 Line 292. Please describe what the introduction (especially the recall period) by the tool for each participant was. At what time, the person should complete the diary etc., after 7 days? This was not clear for me when I started to read the paragraph. Also, wouldn’t it be more appropriate to call it a ‘recall’ rather than a diary? Most people would think that diaries are designed for a continuous monitoring (after each hour, in the evening).

Apologies this was unclear – we have added a sentence (line 317-318) to clarify that users are advised that they would need to report their activity for the preceding 7 days.

We agree that ‘recall’ is a more accurate description, but ‘My Activity Diary’ was purely a user-facing name and one we felt was more accessible and recognisable to potential participants.

Line 305: Would it be possible to list all 83 activities in an online appendix? That would be interesting for other users and researchers designing such tools.

Yes, thank you for this suggestion - we have now created a supplementary file documenting the list of activities included in the DAPPA tool prototype, their associated MET value, and corresponding reference from the Ainsworth compendium. Reference to this appendix has been added in text (Line 321). We should note that we don’t necessarily envisage this being the complete and final list of all activities to be included in the tool – simply that these were the activities we arrived at after this initial development work.

Line 314: Could you describe somewhere how long it takes to fill in the diary for one person, on average? Is it very time consuming? This can also be discussed later, .e.g. with respect to the feasibility of your tool for larger applications.

Many thanks for this suggestion, we agree that this is an important consideration and one we need to include data on in further work. We didn’t collect data on this in the current phase of work as we felt that, at this stage, it wouldn’t provide an accurate sense of how long the tool takes to complete given that we were continuing to adapt and develop it. We did get a sense from the qualitative interviews in the optimisation phase that some participants found the tool quite time intensive to complete and this is something we hope to address by moving to a single day report format.

Line 315: Typo ‘.’

Many thanks - resolved

Line 317: This feedback opportunity is interesting and valuable. Just clarify to me: Do they receive the feedback only once after the activities from the past 7 days were entered? Are there any consequences when you use your tool in intervention studies to measure change in PA by providing cues for changing the target behaviour? Can you de-activate the feedback in the tool?

Yes, participants only receive feedback after all previous 7 days data is entered. We acknowledge that this could potentially have implications for use of the tool in intervention studies as you describe. Accordingly, our plan moving forward is to develop a version of the tool that could be utilised with or without the feedback element. We mention possible development of the feedback element of the tool in our discussion (line 514-520).

Line 320. How many questions for sedentary time were there? Do you cover sedentary time in different domains, sitting at work, sitting during travel, in leisure time for tv viewing, or was it one overall question for daily sedentary time? I remember in the introduction you talked about the importance of different key domains etc.

In the present iteration of the tool, respondents are not asked to directly report all sedentary time, although there are certain activities that users led us to include that have a MET value <1.5. Sedentary time is calculated from these and the time remaining in a 24-hour period after participants sleep duration and all >=1.5 MET activity duration has been taken into account – this is mentioned in lines 362-364 of the manuscript. This approach is something we are keen to validate meticulously, as we believe that, relative to other measures, reporting activity and sleep is much easier than estimating more passive/ habitual sitting time and therefore may lead to better estimations of sedentary time.

 Line 320: Can you provide an example for a tailored feedback message in the text?

Many thanks for this suggestion – an example of the feedback provided is now provided in the text (345-350).

Table 4: See comments Table 1.

Thank you – as in Table 1 we have removed the IMD data and added clarification that all data presented as n(%) unless otherwise specified (line 369).

Table 5: First row. In the updated diary, is there an option to add own/individual activities, if the activity is not listed? Did you consider this option?

This was something we deliberated over, yes, as we shared your view that there may indeed be activities participants do that are not captured by the tool. However, we opted against this because in order to provide equivalent data to other inputs (i.e. the named activities to choose from), the intensity of the activity conducted would also have to be self-judged and reported by the participant, which, as we learned in the planning phase, would likely be highly variable from user to user. We were concerned this would therefore risk the accuracy of the output.

Line 360: I appreciate that you describe the data loss transparently. Could you please explain why the ‘entire’ validation study was deemed as futile? Do you think it is worth showing the mean values, agreement in a descriptive, very brief, way for that 9 people?

Many thanks for the suggestion. We did deliberate over this and have opted not to present the mean values for the validation study for several reasons. The first was that we felt a sample size of 9 would do a disservice to the measurement tools we have used in the study and provide a message that overstates the suitability (or lack thereof) of each one. Secondly, given the loss of data for many of our DAPPA tool responders, our confidence in the data that we did obtain was diminished. We would be much more comfortable ensuring the database is fully functioning before presenting data from the tool in the public domain. Less crucially, knowing that we did not have the majority of this validation data, we chose to focus this paper on the other elements of this development process, and now feel that the brief addition of this data without full and proper discussion may seem like an afterthought. Ultimately, we see this preliminary validation as a pilot to the work we will do to validate the refined tool more rigorously and would refer to it as such in subsequent publication.

 Line 379: This is a good point. That was my impression too. Perhaps using the daily option is promising. It seems to be a lot of work when many participants have to record their activities ‘offline’ on a separate form and then transfer their answers to the digital tool.

We agree - this informed our plans to modify this going forwards as outlined in our discussion – lines 475-481.

 Discussion

 Line 448. I am not sure whether this study (nr 63) really ‘compared’ long and short version, e.g. by evaluating differences between them. Please clarify to me and/or in the text.

Apologies, the language we used was inaccurate – we have amended in the text to clarify to ‘evaluating the relative performance of’(line 478).

Line 470: What about the representativeness of your sample of older adults with respect to the general population? You may elaborate briefly on this topic. Do you have any concerns, as they have been invited using the psychology volunteer participant database; any selection issues with that? Perhaps they have a better understanding of health behaviours and concept? Was that an active or typically inactive sample based on the questionnaire results?

Thanks for raising this point, we have added a brief discussion acknowledging the possibility of selection bias to lines 500-505.

To clarify, the questionnaires were not used to characterise the physical activity of the interview participants – these were simply used as a means to prompt discussion about existing measures of physical activity. However, we know from these focus groups and interview discussions that our sample had a wide range of self-reported activity levels – from those whose limited mobility meant that they only engaged in light activities, small amounts of walking and chair-based activities to those who were regular runners.

Line 475: Just a note. For the future validation study you may consider the accelerometer and, specifically the PASE, a questionnaire which seems to be very appropriate in older adults (doi: 10.1007/s40279-020-01268-x)

Many thanks for your helpful suggestions

Conclusion

Line 500: I suggest adding again a very brief description of the tool to the conclusion section, something like that ‘the tool captures more than 80 activities of the past 7 days and covers different domains (the covered ones) and dimensions (the covered ones) of PA as well SB’.

Thanks for this suggestion – we have added this at lines 542-545.

Related to that could please explain to me again whether your tool covers now ‘all’ domains (i.e., leisure, household, occupation, transport) and dimensions (frequency, duration, type, intensity [light, moderate, vigorous) of PA (as well SB). You may provide details for that in the result/method section too. Readers often search whether the tool measures frequency of PA as well as which types and intensities are assessed. In the introduction I got the impression that your tool should also cover light intensity and the key domains of PA for older adults (what are the key domains again?)

Yes the tool covers all of the domains you mention above, but organised into domains informed by our early development work with regards to how older adults talked about their activity – i.e. ‘home and garden’ – activities in and around the home, ‘out and about’ – activities involved in daily routines and getting from place to place, ‘sport and exercise’ – intentional activity  with the purpose of keeping fit or competitive in nature and active and ‘social and leisure’ – activities that are secondary to socialising with others or hobbies. The tool captures all dimensions of PA – duration, frequency and type are reported, intensity is calculated on the basis of MET values from the Ainsworth compendium. The duration of sedentary behaviour each day is also captured.

We have tried to make these points clearer in our description of the tool – changes made on lines 323-327, and 328-330.

Reviewer 2 Report

The article is devoted to a topic of substantial importance and matches the range of issues generally covered by the International Journal of Environmental Research and Public Health . In terms of formal requirements, the concept of the article is compliant. The authors have employed a method which complies with the criteria to be met by scientific papers. The title of the paper corresponds to the issues it addresses.  The aim of the research is stated clearly. Statistical methods were used apropriatelly. Results were clearly presented in the main text and in tables and figures. Study findings have been interpreted correctly. The authors offer a valid overview of the extent of the problem, building on theoretical concepts and  research. The reference list includes publications that are up to date and well suited to the subject matter of the paper. As regards the formal aspect, the article has been subject to a well-thought-out preparation procedure and conforms to the standards of the International Journal of Environmental Research and Public Health. The conclusions to be drawn from its content are of substantial practical importance and the authors have succeeded in achieving the article’s intended aim.

Author Response

Comments and Suggestions for Authors

The article is devoted to a topic of substantial importance and matches the range of issues generally covered by the International Journal of Environmental Research and Public Health . In terms of formal requirements, the concept of the article is compliant. The authors have employed a method which complies with the criteria to be met by scientific papers. The title of the paper corresponds to the issues it addresses.  The aim of the research is stated clearly. Statistical methods were used apropriatelly. Results were clearly presented in the main text and in tables and figures. Study findings have been interpreted correctly. The authors offer a valid overview of the extent of the problem, building on theoretical concepts and  research. The reference list includes publications that are up to date and well suited to the subject matter of the paper. As regards the formal aspect, the article has been subject to a well-thought-out preparation procedure and conforms to the standards of the International Journal of Environmental Research and Public Health. The conclusions to be drawn from its content are of substantial practical importance and the authors have succeeded in achieving the article’s intended aim.

We thank the reviewer for their time and their positive feedback about our work.

Reviewer 3 Report

1. Introduction
The purpose of the paper is clearly stated.
The problem is well defined.
The case is made that the problem is significant.
The research questions and/or hypotheses are not clearly presented.
The title and abstract are appropriate for the paper.
The introduction is adequate for the paper.

2. Literature review
The literature is recent and relevant.
The literature review acknowledges the depth and breadth of investigation in the field.

3. Methods
The methods used are clearly explained and justified.
The methods are probably sufficient to answer the research questions.

4. Findings
The results are clearly related to the data (including text and numeric).
Figures, tables, and other graphic displays are understandable, and their primary findings are discussed in the text.
The arguments are presented with sufficient logical consistency.

5. Discussion and implications
The findings are not in-depth discussed in relation to the literature review and research questions.
Implications for theory, policy, and practice are not explored fully and elaborated.
Strengths and limitations of the study are not adequately mentioned.

Author Response

Many thanks for your comments - our responses are in bold italics

Comments and Suggestions for Authors

  1. Introduction

The purpose of the paper is clearly stated.

The problem is well defined.

The case is made that the problem is significant.

The research questions and/or hypotheses are not clearly presented.

We acknowledge that we have presented the purpose of the study as research aims as opposed to research questions, but do feel that these are set out clearly. The primary aims of the study are presented in lines 96-101, with further sub aims of the various components of the work now presented on lines 125-128, 151-154 and 177-180. We felt that owing to the qualitative/exploratory nature of this work, it is not appropriate to present hypotheses.

The title and abstract are appropriate for the paper.

The introduction is adequate for the paper.

  1. Literature review

The literature is recent and relevant.

The literature review acknowledges the depth and breadth of investigation in the field.

  1. Methods

The methods used are clearly explained and justified.

The methods are probably sufficient to answer the research questions.

  1. Findings

The results are clearly related to the data (including text and numeric).

Figures, tables, and other graphic displays are understandable, and their primary findings are discussed in the text.

The arguments are presented with sufficient logical consistency.

  1. Discussion and implications

The findings are not in-depth discussed in relation to the literature review and research questions.

We are sorry to read that the reviewer feels that we have not sufficiently discussed our findings.  Throughout the discussion we relate our findings to specific relevant (and often recent) literature e.g. lines 436, 453, 458, 463, 480, 481, 477, 483, 484, 489, 492, 496 and describe how our findings contributed to the development of our tool (i.e. the primary aim of the study). If the reviewer has any specific guidance about how they wish us to improve upon these points, we’d be happy to try and address this.

Implications for theory, policy, and practice are not explored fully and elaborated.

Thank you for suggesting that more attention could be given to the implications of his work within our discussion. Whilst we have not gone into detail about the potential of our tool for use in practice and its implications for older adults and researchers, this was as a result of not wishing to overstate our findings at this stage. This work represents only the first step in developing the tool and we cannot yet claim to have a tool that we know to be appropriate for use in practice. We have however added further text (line 521-529) that suggests what implications the current stage of development and some of the lessons learned along the way may have for wider practice.

Strengths and limitations of the study are not adequately mentioned.

Thank you for this observation. In revising the manuscript in light of another reviewer’s comments we have added to our limitations section (Line 493-510), which we now believe more fully captures the key limitations to this piece of research.